# Impact of Surfactant and Calcium Sulfate Type on Air-Entraining Effectiveness in Concrete

**DOI:** 10.3390/ma15030985

**Published:** 2022-01-27

**Authors:** Maciej Sypek, Rafał Latawiec, Beata Łaźniewska-Piekarczyk, Waldemar Pichór

**Affiliations:** 1Lafarge Cement S.A., 28-366 Małogoszcz, Poland; rafal.latawiec@lafarge.com; 2Department of Building Processes and Building Physics, Faculty of Civil Engineering, Silesian University of Technology, 44-100 Gliwice, Poland; beata.lazniewska-piekarczyk@polsl.pl; 3Department of Building Materials Technology, Faculty of Materials Science and Ceramics, AGH University of Science and Technology, 30-059 Krakow, Poland

**Keywords:** air-entrained concrete, surfactant, air-entrainment mechanism, admixture adsorption

## Abstract

The paper presents the evaluation of the influence of calcium sulfate on the air void microstructure in concrete and its action mechanism depending on the character of the air-entraining agent. Gypsum dehydration has been previously proven to negatively influence the air void structure of air-entrained concrete. Ettringite, nucleating from tricalcium aluminate and calcium sulfate, influences the adsorption and mode of action of anionic-based polycarboxylate ether admixtures. The authors suspected the admixture’s air-entraining mechanism was also affected by these characteristics. Gypsum dehydration was confirmed to influence the air void structure. In the case of the anionic surfactant, the content of air bubbles smaller than 300 µm was lower compared to cement with gypsum and hemihydrate. On the other hand, the content of air voids with a diameter up to 60 µm, which are the most favorable, was higher. The results obtained led to the conclusion that the mechanism of air entrainment was twofold, and in most cases occurred through the lowering of surface tension and/or through the adsorption of surfactant on cement grains. The adsorptive mechanism was proved to be more effective in terms of the total air content and the structure of the air void system. The results and conclusions of the study provide guidelines to determine the proper surfactant type to reduce the risk of improper air entrainment of concrete, and emphasize the importance of gypsum dehydration of cement in the process of air entrainment.

## 1. Introduction

### 1.1. Cement Hydration

Differences in the chemical and phase composition of cement affect the pore structure of the concrete [1]. Among chemical properties, the C_3_A content and crystal composition, alkali content, type, and content of the sulfate-bearing material have been pointed out as the most important in previous studies [2,3,4,5,6]. The above-mentioned properties play an important role in cement hydration, especially in the preinduction and induction stages [7]. Portland cement consists of polymineralic grains mixed with different forms of calcium sulfate. Each grain consists of various phases, mostly: tricalcium silicates, dicalcium silicates, calcium aluminates, and ferrites. As is well known, the hydration of Portland cement is time-dependent, and four major stages can be distinguished [8]:−Initial period—mainly dissolution of calcium aluminates and tricalcium silicates, and precipitation of calcium sulfoaluminate hydrates;−Dormant period—hydration rate decreases;−Main hydration—acceleration of the dissolution of dominant silicate-rich phases and precipitation of calcium silicate hydrates and calcium hydroxide lead to setting and early strength development of the cement suspension;−Continuous hydration—strength development.

Hydration typically starts with the dissolution of calcium aluminate and sulfate, then tricalcium silicate and ferrite, and finally dicalcium silicate. Ions, mostly calcium (Ca^2+^), sulfate (SO_4_^2−^), sodium (Na^+^), potassium (K^+^), and hydroxide (OH^−^), are released to the pore solution, and ion exchange follows. On the surfaces of different phases, the densification of hydration products results in the formation of more and more impermeable layers [9]. After the phases are covered with products, the hydration is controlled by ion diffusion through the product layer. Aluminate hydration products are AF_t_ and AF_m_, while C_3_S hydration products are mainly calcium silicate hydrate (C-S-H) and calcium hydrate (CH) [7,10]. Both AF_t_ and AF_m_ are large groups of compounds. However, their most common representatives are ettringite and monosulfate. Monosulfate forms when the concentration of sulfates in the pore solution is low due to, for example, gypsum exhaustion. Ettringite occurs in needle- or rodlike structures, while monosulfate shows an amorphous structure or, less often, a layer or plateletlike structure. C-S-H nucleates in spherical forms, and later transforms into a network [7].

### 1.2. Role of Calcium Sulfate in Cement Hydration

Many factors can influence the hydration rate of each of the phases. The initial rate of C_3_A hydration is strongly influenced by the source of sulfate used, therefore it should be considered together with the type of sulfate-bearing material [11,12]. Porchet, when studying the early hydration of C_3_A in the presence of different types of calcium sulfate, showed that compared to gypsum, hemihydrate, due to its higher solubility and solubility rate, provided a higher sulfate ion concentration in the pore solution [10]. Many hypotheses concerning the mechanism of calcium sulfate action have been formulated; however, the mechanisms of C_3_A reaction retardation are not fully elucidated yet. According to some authors, it occurs because the hydration products are forming a more or less permeable sulfoaluminate crystalline or noncrystalline barrier layer on the surface of C_3_A, limiting the transport of water and ions [13]. The other hypothesis is that the deceleration of C_3_A hydration is related to the precipitation of calcium hydroaluminate or the adsorption of sulfates on C_3_A [14]. Such disagreements in interpretations are probably because most of the phenomena occur at a very early stage, even during mixing. Łagosz showed that the effectiveness of C_3_A reaction retardation in the presence of hemihydrate as lower than when gypsum was used [15]. Additionally, the product of C_3_A and hemihydrate reaction is not analogous to the ettringite phase; thus, the monosulfoaluminate form of C_3_A∙CaSO_4_∙11H_2_O does not undergo the ettringite–monosulfate reaction. Moreover, if the cement contains a low amount of C_3_A, the increased concentration of ions in the pore solution will have a greater effect, as there is less C_3_A to adsorb the sulfate ions

### 1.3. Chemical Admixtures Adsorption Mechanisms

Cement grains, covered with a “membrane” of products and mixed with water, are a colloid that forms an electrical double layer due to the product’s high surface charge density and the highly ionic character of the pore solution [16,17,18]. Stern’s electrical double layer model implies the existence of the Stern layer—a layer of ions that are of the opposite charge to the solids. Ions contained in the Stern layer are bound to the surface of solids, while those in the diffuse double layer are not. The dividing line between the Stern layer and the diffuse double layer is called the shear plane. The electrokinetic potential measured at the shear plane is called the zeta potential [17].

Polymineral cement grains consist of different phases. Thus, oppositely charged areas may exist on the surface of a single cement grain. The zeta potential of cement’s early hydration products, especially C-S-H, C_3_S, and AF_t_, has been reported as both negatively and positively charged, depending mostly on the chosen chemical composition of the solutions [8,19]. Zingg [8] measured the charge of C_3_S and C-S-H to be positive and of ettringite to be negative, which corresponded to the data obtained by Planck [19]. The author also pointed out that charge inversion occurred in a specific chemical environment, similar to cement’s pore solution, due to the adsorption of oppositely charged ions on the Stern layer of the solid. However, not all ions are equally efficient at charge inversion.

Planck [19] proved that the presence of ettringite is necessary to achieve a high PCE superplasticizer adsorption. Zingg [8] proved that PCE adsorption directly on ettringite is needed to achieve good superplasticizer effectiveness.

### 1.4. Theories of Air-Entraining Agents Adsorption Mechanism

Air-entraining agents are usually surfactants—molecules composed of a hydrophilic head and a hydrophobic hydrocarbon tail. The hydrophilic head can be either charged (ionic) or polar (nonionic).

Surfactants enhance foam creation and stability through several mechanisms; namely, by lowering surface tension, separating bubbles from each other by steric interactions, and stabilizing the bubbles through micellization [20,21,22].

Theories about the action mechanism of AEA:Folliard and Du stated that air-entraining agents act mainly through the portion of the admixture concentrated at the liquid–air interface, while the portion present in the bulk liquid phase serves as a reserve. The portion adsorbed on solids contributes little to air entrainment [23].Zhang et al. concluded that anionic surfactants adsorb on cement grains with the hydrophobic chain oriented toward the bulk phase. They pointed out that cement grains have a negative charge, which was later proved to differ regarding specific products (AF_t_ or C-S-H/C_3_S) [24].Qiao et al. studied the adsorption of different types of surfactants on cement particles, and proved that they adsorb on oppositely charged solid surfaces. The presence of salts in the solution, namely Ca^2+^ ions, was determined to weaken the surface activity of anionic surfactants. However, only the total air content was tested, not the air void structure [25].Petit et al. showed that, to maintain high foam stability, it is necessary to obtain monolayer adsorption of the surfactant on the solid’s surface [26].Liu et al. studied the air void parameters of mortars and the adsorption of different types of surfactants on cement grains [27]. The presented explanation was contrary to the aforementioned results obtained by Planck and Zingg [8,19].Sahin et al. and Tunstall et al. proposed surface tension measurement as a useful method to determine the adequate amount of AEA required to obtain a sufficient air void system [28,29]. On the other hand, Huang et al. proved that surface tension is not the decisive factor affecting foamability [30].Shan et al. proposed a coworking model between the anionic and nonionic surfactants, positively affecting the air–water interface [31].

In this study, the relationship between the calcium sulfate type, surfactant type, and the air void structure of air-entrained concrete was examined together with the hydration behavior of such systems. It was observed that gypsum dehydration had an effect on the air void structure when using anionic surfactants, and it was concluded that using another type of surfactant may prevent such behavior. Moreover, it was proven that, among the tested surfactant types, the anionic surfactant gave the best air void structure. It was observed that the interaction between surfactants and cement differed depending on the type of surfactant. A hypothesis was proposed on the action mechanism of anionic surfactants, which is critical to the previously mentioned state of the art.

## 2. Materials and Methods

### 2.1. Experimental Program

The influence of the type of calcium-sulfate-bearing material present in cement and different types of surfactants on the air void structure was examined in three stages:Cement mortar—examination of foaming behavior:The first stage was performed on cement mortars obtained with the use of cements with different calcium-sulfate-bearing materials and different surfactants at a constant dosage. This study aimed to test the air-entraining abilities of different surfactants in reaction with different sulfate-bearing materials.Concrete mixes—analysis of the air void structure of fresh and hardened concretes:The second stage was conducted on fresh and hardened concretes using the surfactant as the only admixture. The air content was kept at a constant level regardless of the surfactant type. The type of surfactant and sulfate-bearing material and the admixture content were the variables. The aim of this study was to examine the influence of different surfactants and sulfate-bearing materials on the air void parameters of concrete.Pore solution—ion content in pore solution, surface tension, and total organic carbon (TOC):The third stage was performed on pore solution samples prepared from cement pastes. This study aimed to discover the specific interactions between cement particles and surfactants that lead to different air void structures in concrete mixes. Similar to the previous stages, the surfactant dosage was fixed to ensure an identical content of the active substance in all the analyzed cases.

The selected test methods and materials used in the research have been elaborated based on Lafarge’s pragmatic experience.

### 2.2. Materials

Cements with different types of sulfate-bearing materials were prepared for all stages of the experimental program. One batch of clinker characterized by a low C_3_A content was used to obtain CEM I 42.5 N-SR3/NA-type cements. Cement with gypsum only was prepared by intergrinding clinker with pure flue gas desulfurization (FGD) gypsum. Cement with hemihydrate was prepared through the controlled heating of cement with gypsum in an oven at 90 °C for 15 h. Cement with anhydrite addition was prepared by intergrinding clinker together with FGD gypsum and anhydrite, similarly to the cement with gypsum only.

The phase composition of the clinker, the content of sulfate-bearing materials, and the calculated level of gypsum dehydration (GDH) are presented in Table 1. The physical properties of the types of cement are presented in Table 2. The cements were named based on the type of sulfate-bearing material. The abbreviations used for the cement type are: CGG—cement with predominant gypsum content; CGH—cement with a significant amount of hemihydrate; and CGA—cement with the addition of anhydrite.

Cement with hemihydrate was heated in metal containers with a capacity of 20 kg. Following heat treatment, the cement samples were left to cool down. This process simulated the possible dehydration occurring during milling or storage in silos. The cement was then homogenized by mixing. The gypsum dehydration level (GDH) was calculated as follows:(1)$$\mathrm{GDH}=\left(\mathrm{HH}\cdot \frac{172}{145}\right)/\left(G+\mathrm{HH}\;\cdot \frac{172}{145}\right)$$
where the molar mass of gypsum (G) was 172 g/mol, and the mass of hemihydrate (HH)—145 g/mol. The content of Na_2_O and K_2_O was approx. 0.2% and 0.5%, respectively.

River sand and 2/8 and 8/16 fractions of crushed amphibolite were used. All aggregates met the requirements of the PN-EN 12620 standard, and were proven to be frost-resistant. The water absorption of river sand, 2/8 amphibolite, and 8/16 amphibolite was 0.5%, 0.9%, and 0.9%, respectively. 

Pure liquid surfactants of different types were used. The concentration of the active substance in the surfactants was fixed at 2% in all cases. All the admixtures used are described in Table 3.

### 2.3. Preparation of Samples

All mortars were mixed according to the standard procedure described in PN-EN 196-1 [32]. During the preparation of mixtures and the following tests, a temperature of 20 ± 2 °C and a relative humidity greater than or equal to 50% were maintained in the laboratory. The surfactants were added to water prior to mixing with cement. The surfactant content was fixed at 0.5% in relation to the mass of cement.

The aggregates were dried in an oven and placed in a temperature-controlled room at 20 °C for at least 24 h before mixing. At the beginning of mixing, all the aggregates were loaded into the mixer along with approximately half of the mixing water, and subsequently mixed for 20 s in order to bring them to the saturated surface dry (SSD) state and ensure that they were evenly distributed in the material. Afterward, the cement and remaining water were added and further mixed for 15 s. Then, the surfactant was added and the concrete was mixed for one more minute. The resulting mixture was rested for 5 min and then mixed for an additional 30 s. The concretes were tested for slump, total air content, and air void structure, as measured by the air void analysis (AVA) method. Samples were formed for hardened concrete air void analysis.

Filtration under vacuum, also known as Büchner filtration, was used to isolate the cement paste precipitate from the solution (Figure 1).

The pore solution was vacuum-extracted from the prepared cement pastes, characterized by a water-to-cement ratio of 0.45 and an air-entraining admixture content of 0.5% in relation to the mass of cement. Cement pastes were prepared in accordance with the PN-EN 196-3 [34] standard procedure. Pastes were filtered through a Büchner funnel 10 min after mixing, which was carried out in an automated laboratory mortar and paste mixer. The temperature of both the cement paste and ambient air was 20 °C.

### 2.4. Test Methods

The consistency of the fresh cement mortars was tested with the use of equipment compliant with PN-EN 1015-3 [35]. The cement mortar air content was measured using a setup described in the PN-EN 1015-7 standard [36].

The concrete air content was determined using the pressure method, according to PN-EN 12350-7 [37]; slump was measured according to PN-EN 12350-2 [38]. The air void characteristics in the fresh concrete were determined using an AVA-3000 Air Void Analyzer [39]. The estimated AVA method’s error was 0.014 mm for the spacing factor and 0.12% for A_300_. The error estimate was based on repeated measurements of a few batches of identical concrete.

The air void structure in the hardened concrete was determined in accordance with PN-EN 480-11 [40]. The analysis was performed on a RapidAir apparatus, dedicated to this test and compliant with the guidelines of the standard.

In the second stage of the study, nine concrete mixes with the following compositions were prepared: modified cement—350 kg; aggregates: fine—640 kg, 2/8 mm (amphibolite)—496 kg, 8/16 mm (amphibolite)—637 kg; w/c ratio—0.48. The admixture dosage was adjusted to obtain an air content of 5.5 ± 0.5% after 60 min. The target concrete slump was 150–200 mm (Table 4).

The sulfate (SO_4_^2−^) concentration was determined by the gravimetric method using barium chloride, in accordance with PN-ISO 9280 [41]. The method is based on determining the mass of barium sulfate precipitate formed in the reaction of barium ions with sulfates present in the sample. During the procedure, sulfates were precipitated from hot solution with barium chloride, forming sparingly soluble barium sulfate. After filtration and calcination at 815 °C, the precipitate was weighed, and the sulfate content was calculated from the amount of barium sulfate, according to Formula (2) below:(2)X=m·0.4114/V
where X is the sulfate content (mg/dm^3^), m is the mass of the sample after calcination at 815 °C (mg), 0.4114 is the conversion factor from BaSO_4_ to SO_4_^2−^, and V is the volume of the solution used in the procedure (dm^3^).

The sodium, calcium, and potassium contents were determined by flame atomic emission spectrometry, according to PN-ISO 9964-3 [42]. The basis of the qualitative analysis in this method is the wavelength of the spectral line and the image of the spectrum. Flame photometry is widely used for the determination of alkali and alkaline earth metals content. The aqueous solution obtained is introduced into an air–acetylene flame, in which the potassium, sodium, calcium, and lithium atoms are excited and begin to emit light at a characteristic wavelength. The radiation intensity was proportional to the concentration of the elements in the tested sample. The measurement was conducted using a flame photometer.

The total concentration of inorganic and organic carbon (TIC/TOC) in the pore solution samples was determined using a Shimadzu TOC-L analyzer (Kyoto, Japan). The apparatus uses the combustion catalytic oxidation method at a temperature of 680 °C. On the other hand, the measurement of the concentration of carbon dioxide formed in the process was carried out with a nondispersive infrared sensor (NDIR). This allowed the efficient oxidization of not only low-molecular-weight, easily decomposed organic compounds, but also hard-to-decompose insoluble and macromolecular organic compounds. The analyses were carried out on the basis of the concentration curves of total carbon (TC) and inorganic carbon (IC), ranging from 0 to 100 mg/dm^3^. This allowed the IC and TC values to be measured directly, while the TOC value was determined by subtracting the IC value from the TC value.

The measurements of the surface tension coefficient of pore solutions extracted from cement pastes were carried out with the use of a stalagmometer. The basic element of this instrument is a thick-walled glass capillary with an internal diameter selected to force the tested fluid to run in drops. The measurement consists of counting the number of drops of both the reference and tested liquids falling from the stalagmometer during the flow of a fixed volume of liquid. Deionized water was used as the reference liquid due to its availability and well-described physical properties. The stalagmometer was rinsed with the tested liquid prior to the measurements. Five repetitions were carried out for both the reference liquid and the pore solution samples. The surface tension coefficient of the tested liquid (σ_p_) was determined on the basis of Equation (3):(3)$$\sigma_{p^{-}}=\sigma_{H2O}\cdot n_{H2O}\cdot \rho_{p}/\left(n_{p}\cdot \rho_{H2O}\right)$$
where σ_p_ is the surface tension of the sample (N/m), σ_H2O_ is the surface tension of water, n is the measured number of drops of water (H_2_O) and the tested sample (p), and ρ is the density of the sample (p) and water (H_2_O) at the measurement temperature. The density of the tested samples was determined with a pycnometer at ambient temperature. The water density at the measurement temperature was determined using Equation (4):(4)$$\rho =5.459\cdot M/\left(0.3054^{1+{\left(1-\frac{T}{647.13}\right)}^{0.081}}\right)$$
where ρ is the density of water (kg/dm^3^), M is the molar mass of water (18.01528 g/mol), and T is the temperature (K). 

The water surface tension coefficient at the measurement temperature was determined using correlated experimental data from the Dortmund Database [43].

## 3. Results

### 3.1. Characteristics of Mortars

All the results (presented in Table 5) proved that the use of different sulfate-bearing materials had little to no effect on the total air content of the cement mortars. However, the type of surfactant had a significant influence. The surfactant dosage was fixed at 0.5% in relation to the mass of cement, regardless of the type of surfactant.

All the surfactants used had the ability to entrain air. However, their foamabilities were different. The anionic surfactant entrained the highest amount of air, followed by the nonionic and cationic surfactants.

### 3.2. Air Void Parameters of Concrete

The air void parameters of the fresh concrete mixes are presented in Figure 2, Figure 3 and Figure 4.

In the case of anionic surfactant, the impact of calcium sulfate type was moderate: lower on spacing factor and higher on A_300_. Gypsum provided the best air microstructure, due to the lowest spacing factor and highest A_300_. Hemihydrate and anhydrite presented a similar air system. 

In the case of the nonionic surfactant, the impact of the calcium sulfate type was high, both on spacing factor and on A_300_. Gypsum provided the best air microstructure, due to the lowest spacing factor and highest A_300_, followed by hemihydrate and anhydrite.

In the case of cationic surfactant, the impact of the calcium sulfate type was high, both on spacing factor and on A_300_. Gypsum provided the worst air microstructure, due to the highest spacing factor and lowest A_300_. 

The hardened concretes’ air void parameters are presented in Figure 5, Figure 6 and Figure 7. The results were grouped to provide a clear view of the influence of the type of sulfate-bearing material on the air void structure. In the enlarged area, the data were narrowed to the air void diameter range of 0–60 µm due to the fact that such air voids are the most favorable in terms of freeze–thaw resistance, according to Łukowski [44].

In the case of the anionic surfactant, the content of air voids with a diameter up to 300 µm was the highest for pure gypsum and the gypsum/anhydrite mix, while the hemihydrate presented the lowest A_300_. On the other hand, gypsum provided the lowest amount of air voids with a diameter up to 60 µm compared to both anhydrite and hemihydrate. The differences will be discussed in a subsequent section. The data obtained with AVA were slightly different—the A_300_ content was the highest in the case of gypsum, followed by equal results obtained for hemihydrate and anhydrite.

The nonionic surfactant provided a worse air void structure than the anionic surfactant, according to the results of the hardened concrete air void system analysis and the AVA measurements. The correlation between the results obtained for hardened and fresh concrete was poor. According to the hardened concrete air void structure analysis, the differences between cements were clearly noticeable; gypsum presented the lowest amount of air voids with a diameter up to 60 µm, similarly to the anionic surfactant.

Cationic surfactant provided the poorest air void structure compared to the other admixtures. Variations in the composition of sulfate-bearing materials had less influence on the air void structure than in the case of the anionic and nonionic surfactants.

### 3.3. Pore Solution—Ion Contents, Surface Tension, TOC

The results of the pore solution analysis are presented in Table 6 and Figure 8.

As anticipated, the type of sulfate-bearing material had an influence on the ionic composition of the pore solution. Hemihydrate presented higher Ca^2+^/SO_4_^2−^ and lower Na^+^/K^+^ concentrations than gypsum. However, the addition of anhydrite, which is less soluble than gypsum or hemihydrate, yielded Ca^2+^ and SO_4_^2−^ concentrations higher than in the case of gypsum. Overall, the pore solution composition of cement with anhydrite was similar to cement with hemihydrate. The TOC in the case of anionic surfactant AEA-1 was the lowest, and showed a high level of adsorption, while the surface tension was comparable to the solution with no AEA. The TOC of the nonionic AEA-2 and cationic AEA-3 surfactants was high, and showed a low level of adsorption, while for AEA-2, the surface tension was significantly reduced.

## 4. Discussion

As can be seen in Table 5, different types of surfactants showed significantly different foamabilities. The anionic surfactant entrained the highest amount of air, followed by the nonionic and cationic surfactants. Additionally, the nonionic surfactant was characterized by a wetting ability, increasing the consistency of the mortar. The air void structure analysis performed on fresh and hardened concretes returned slightly different results. This effect was attributed to the imperfections of the AVA method. The results obtained for hardened concrete provided an insight into the actual structure of the air void system generated within the concrete.

The air void structure analysis shows that the anionic surfactant provided the finest air voids in both the fresh and hardened concretes, followed by the nonionic and cationic surfactants.

The influence of gypsum dehydration on the structure of the air void system has already been proved in earlier papers [45]. The same results were obtained in the current study—gypsum dehydration caused a decrease in the A_300_ content, while the total air content of the concrete mixtures was unchanged. However, in the previous study, only A_300_ was investigated. Recent results raise questions regarding whether it is better for the freeze–thaw resistance of concrete to produce a higher A_300_ content and a lower amount of microvoids up to 60 µm, or a lower A_300_ but with more microvoids.

The TOC results showed a clear difference in the adsorption of different types of surfactants. The low TOC in the case of the anionic surfactant resulted from its strong adsorption on cement grains. Accordingly, no significant decrease in surface tension was observed due to the fact that the surfactant was not present in the pore solution. The high TOC of the pore solution and up to 25% lower surface tension proved that the nonionic surfactant showed little adsorption of cement grains and remained in the pore solution. In the case of the cationic surfactant, both the TOC and surface tension results were similar to those obtained for the nonionic surfactant, which pointed toward the conclusion that the cationic surfactant was characterized by a similar adsorptive behavior.

The results of the pore solution and air void system analyses led to a hypothesis regarding the mechanism of air entrainment. The literature states that air entrainment occurs through one of two mechanisms: the surfactant adsorption on cement particles or the lowering of surface tension. This study proved that the mechanism depends on the surfactant type. Anionic surfactants were adsorbed on cement particles due to their negative charge and the positive surface charge of ettringite. Ettringite proved to be crucial in the anionic PCE superplasticizer adsorption despite its low content in relation to the other hydration products [8,19]. Those surfactant particles then stabilized and captured air entrained during mixing. On the other hand, the nonionic and cationic surfactants did not adsorb on cement particles, but rather stayed in the pore solution, lowering the surface tension and thus entraining air on the air–liquid interface only. The first hypothesis lies contrary to the observations made by Folliard and Du [23], that the adsorbed admixture did not contribute to air entrainment. However, multiple studies presented in the introduction confirmed such behavior, which points toward the conclusion that the mechanism is complex and difficult to generalize.

The results of the air void system analysis of concrete mixtures made with different surfactants point toward the conclusion that air entrainment through the adsorption on cement particles was superior to the other mechanism in terms of foamability and air void structure—the admixture dosage was several times lower, and smaller air voids were generated.

Pore solution examinations confirmed the earlier hypothesis that gypsum dehydration would result in higher Ca^2+^ and SO_4_^2−^ ion concentration in the pore solution. The concentration was up to 10% higher. The pore solution was also characterized by a lower alkali ion content.

In the case of anhydrite addition as a part of the sulfate-bearing material, the results were unexpected. Anhydrite is characterized by a lower solubility and solubility rate than gypsum, and thus its addition should result in a lower calcium and sulfate ion concentration in the pore solution. However, the studies showed that their concentration was slightly higher than in the case of gypsum. The authors associated such behavior with the cement preparation method. Cement with hemihydrate was obtained from the method with pure gypsum through controlled heating in an oven. Therefore, their particle size distribution did not vary. However, anhydrite was milled together with clinker and gypsum. The author’s unpublished results directly showed that the anhydrite was ground much more easily than gypsum. Thus, the anhydrite part of the sulfate-bearing material should be finer, and its effective solubility rate can be higher than that of coarser gypsum. Similarly to hemihydrate, the alkali ion concentrations were lower than in the case of gypsum. 

There was no difference in both the TOC and surface tension of cements with different sulfate-bearing materials. Thus, gypsum dehydration did not directly influence either the total organic carbon content or the surface tension.

Gypsum dehydration had an effect on the air void structure. In the case of the anionic surfactant, the A_300_ content was up to 18% lower compared to cement with gypsum and hemihydrate. On the other hand, the content of air voids with a diameter up to 60 µm, which are the most favorable, was up to 170% higher. 

The addition of anhydrite resulted in a relatively high content of both A_300_ and air voids up to 60 µm. However, due to the previously mentioned cogrinding of cement with gypsum and anhydrite, the direct influence of the anhydrite’s solubility rate on the air void structure could not be determined.

Qiao et al. [25] previously identified the influence of calcium ions on the surface activity of anionic surfactants. However, no such behavior in terms of the total air content of mortars was observed, even though the calcium concentration among the different cements varied by up to 10%.

Based on the proposed hypothesis regarding the mechanism of air entrainment, gypsum dehydration influenced the air void structure by altering the hydration of cement (especially C_3_A) and the formation of ettringite, which was characterized by the highest adsorptive potential. Variations in the calcium ions content in pore solutions containing different types of calcium sulfate were proven not to alter the air void structure.

## 5. Conclusions

Based on the results obtained in the study, the following conclusions can be formulated:−The results of the pore solution analysis proved that air entrainment in concrete could be achieved through one of two mechanisms: the adsorption of anionic surfactant on the ettringite surface or the lowering of surface tension. −The anionic surfactant was characterized by a better foamability and provided a better structure of the air void system; thus, the adsorptive mechanism of air entrainment can be pointed out as being more effective.−Gypsum dehydration has been proven to alter the air void structure of concrete. Due to gypsum dehydration, the content of air voids with a diameter up to 300 µm decreased by approximately 18% when the anionic surfactant was used. However, the content of air voids with a diameter up to 60 µm, which are the most favorable in terms of the freeze-thaw durability, was higher. −Gypsum dehydration influenced the air void structure by altering the hydration of C_3_A and the formation of ettringite, rather than affecting the surfactant’s mode of action directly.

## Figures and Tables

**Figure 1 materials-15-00985-f001:**
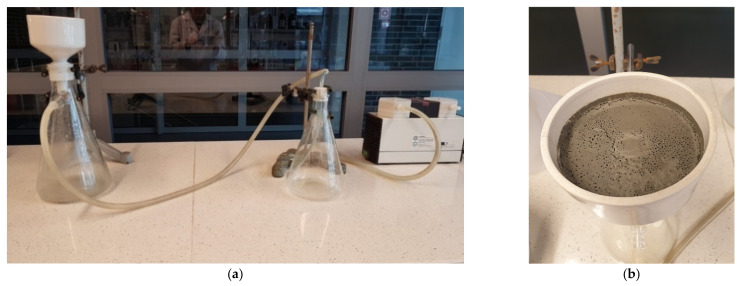
Vacuum filtration of cement paste through a Büchner funnel [33] ((**a**) shows filtration equipment, (**b**) shows cement paste after filtration).

**Figure 2 materials-15-00985-f002:**
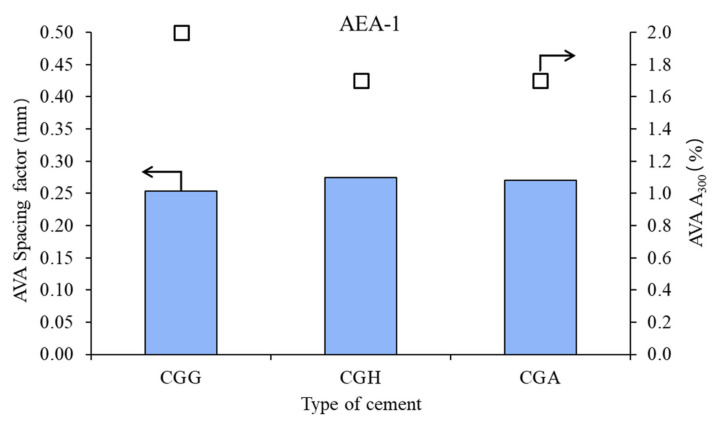
Air void parameters of fresh concrete measured by AVA in the case of anionic admixture AEA-1 (bars—spacing factors; points—A_300_).

**Figure 3 materials-15-00985-f003:**
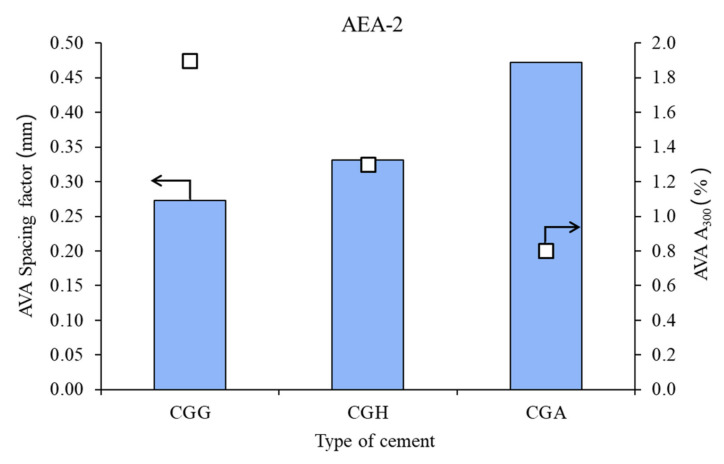
Air void parameters of fresh concrete measured by AVA in the case of nonionic admixture AEA-2 (bars—spacing factors; points—A_300_).

**Figure 4 materials-15-00985-f004:**
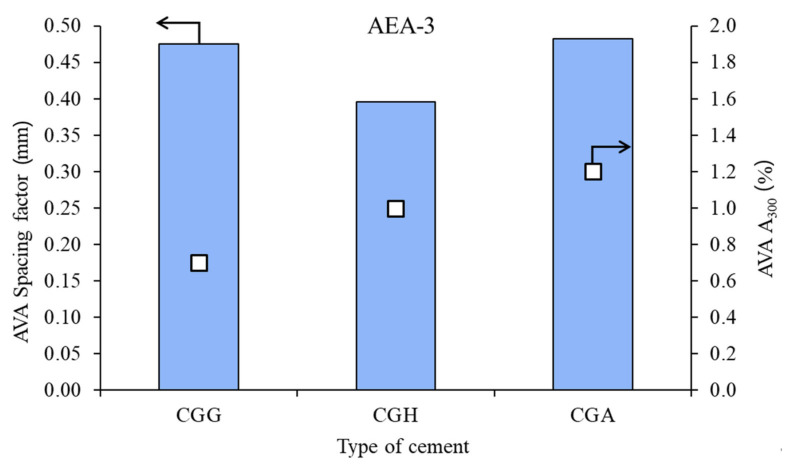
Air void parameters of fresh concrete measured by AVA in the case of cationic admixture AEA-3 (bars—spacing factors; points—A_300_).

**Figure 5 materials-15-00985-f005:**
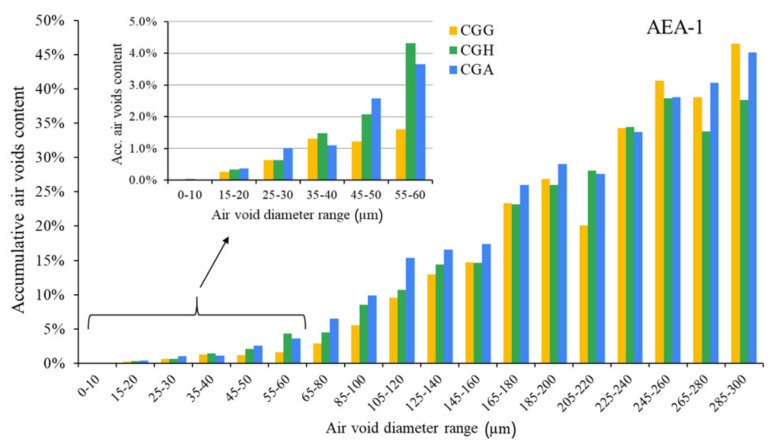
Air void parameters of hardened concrete in the case of AEA-1 (CGG—cement with high gypsum content, GDH = 0.11; CGH—cement high hemihydrate content, GDH = 0.37; CGA—cement with anhydrite addition, GDH = 0.17).

**Figure 6 materials-15-00985-f006:**
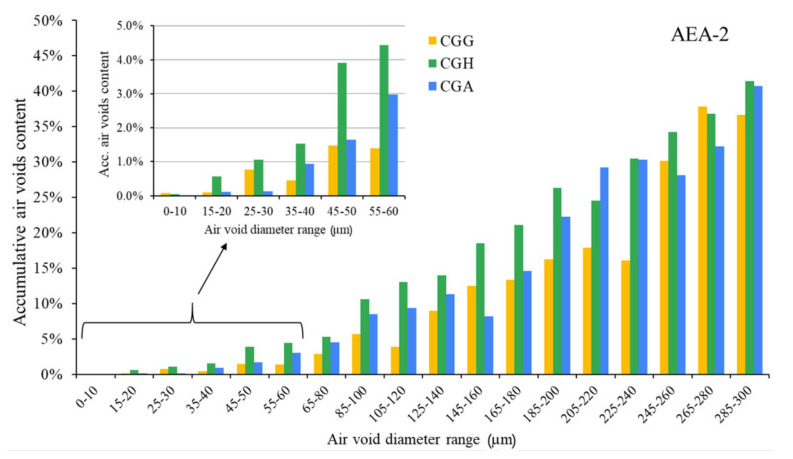
Air void parameters of hardened concrete in the case of AEA-2 (CGG—cement with high gypsum content, GDH = 0.11; CGH—cement high hemihydrate content, GDH = 0.37; CGA—cement with anhydrite addition, GDH = 0.17).

**Figure 7 materials-15-00985-f007:**
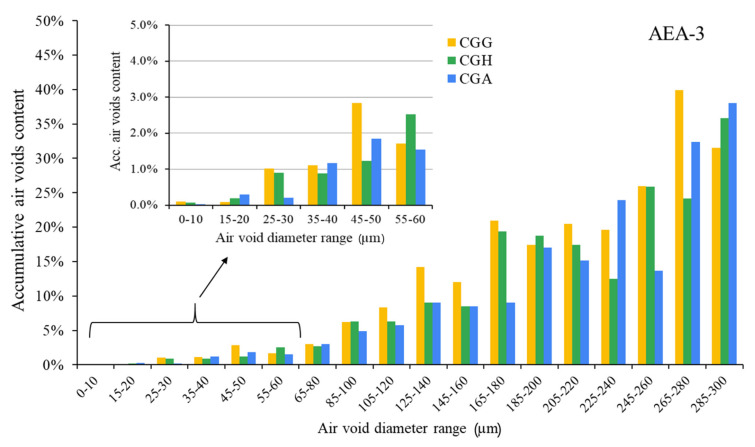
Air void parameters of hardened concrete in the case of AEA-3 (CGG—cement with high gypsum content, GDH = 0.11; CGH—cement high hemihydrate content, GDH = 0.37; CGA—cement with anhydrite addition, GDH = 0.17).

**Figure 8 materials-15-00985-f008:**
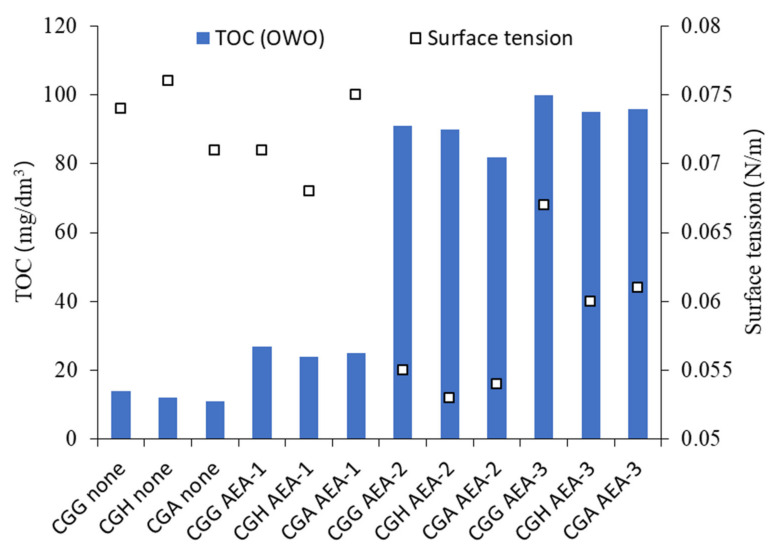
TOC and surface tension of pore solution depending on cement and admixture type.

**Table 1 materials-15-00985-t001:** Cement composition and calculated gypsum dehydration level (GDH), %.

Cement Type	C_3_S, %	C_2_S, %	C_4_AF, %	C_3_A, %	CaO, %	G *, %	HH **, %	A ***, %	GDH	SO_3_, %
CGG	56.7	18.3	16.9	1.5	0.7	2.9	0.3	0	0.11	2.16
CGH	58.4	18.7	17.4	1.3	0.7	1.6	0.8	0	0.37	2.07
CGA	56.7	18.3	16.9	1.5	0.7	1.7	0.3	0.8	0.17	2.27

* Gypsum; ** hemihydrate; *** anhydrite.

**Table 2 materials-15-00985-t002:** Physical properties of cements.

Cement Type	Fineness, cm^2^/g	Soundness, mm	Standard Consistency, %	Initial Setting Time, min	Final Setting Time, min	Compressive Strength, MPa
CGG	3245	0.5	27.6	255	375	50.2
CGH	3112	0.5	28	295	445	52.8
CGA	3432	0.5	27	210	305	52.2

**Table 3 materials-15-00985-t003:** Admixture information.

Surfactant Name	Surfactant Type	Admixture Base	INCI Name
AEA-1	Anionic surfactant	Alcohols, C_12–14_, ethoxylated (1–2.5 EO), sulfated, sodium salts	Sodium laureth sulphate
AEA-2	Nonionic surfactant	Alcohols, C_13_, branched, ethoxylated	Isotrideceth-12
AEA-3	Cationic surfactant	Quaternized and ethoxylated fatty amine	PEG-15 cocomonium methosulfate

**Table 4 materials-15-00985-t004:** Concrete mix proportions.

Cement Type	Surfactant Type	Surfactant Dosage, % in Relation to Cement Mass	Air Content after 60 min, %	Slump after 60 min, mm
CGG	AEA-1	0.5	5.4	170
CGH		0.5	5.5	160
CGA		0.5	5.1	160
CGG	AEA-2	2.7	5.8	190
CGH		2.7	5.4	200
CGA		2.7	5.1	190
CGG	AEA-3	4.2	5.1	190
CGH		4.2	5.8	170
CGA		4.2	5.5	190

**Table 5 materials-15-00985-t005:** Properties of fresh mortars.

Cement Type	No Admixture		AEA-1		AEA-2		AEA-3	
	Consistency, mm	Air Content, %	Consistency, mm	Air Content, %	Consistency, mm	Air Content, %	Consistency, mm	Air Content, %
CGG	159	4.0	174	18.5	196	10.3	169	5.8
CGH	159	3.8	173	18.5	191	9.8	174	6.0
CGA	162	3.7	177	18.5	197	10.7	171	5.8

**Table 6 materials-15-00985-t006:** Ion contents in pore solution depending on cement type.

Cement Type	Admixture	Na^+^, mg/dm^3^	K^+^, mg/dm^3^	Ca^2+^, mg/dm^3^	SO_4_^2−^, mg/dm^3^
CGG	none	416	4039	567	5807
CGH	none	336	3638	675	6520
CGA	none	331	3530	655	6010

## Data Availability

The data are contained within the article. Additional data are available upon request from the corresponding author.

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
