# Peer review of "Impact of Surfactant and Calcium Sulfate Type on Air-Entraining Effectiveness in Concrete"

_materials, 2022, doi:10.3390/ma15030985_

Round 1

Reviewer 1 Report

  1. The abstract mentioned that the results obtained lead to the conclusion that the mechanism of air entrainment is complex. There is no new findings from this investigation, since the complex effects can already be explained by previous research.
  2. The conclusion of this paper should be revised in a list, list the main conclusions that related to the experimental results and discussion.
  3. The contribution of this paper should be pointed out more clearly.

Author Response

Dear Reviewer,

Please find the responses and manuscript after your remarks and suggestions.

Thanks a lot for your valuable input to the manuscript

kind regards

Maciej Sypek

Reviewer 2 Report

The present paper is a study the air entrainment effectiveness in concrete with regard to the type and content of surfactant and the type of calcium sulfate in cement. The paper is interesting for publication. However, it needs to correct some points as presented below : 

  1. Acronyms like PCE, C3A, C-S-H, TOC and others should be expand at at the first time appearance in the manuscript and subsequent contents can use acronyms.
  2. Font style in figures labelling and body of text in the paper is contradicting so authors are suggested to go through author guidelines from the journal and correct accordingly.

Author Response

(The authors gave the same response as above.)

Reviewer 3 Report

Dear Authors,
Thank you for the opportunity to review this article.
The manuscript deals with the evaluation of the effect of calcium sulfate on the microstructure of air voids in concrete. 
The results of your research are definitely interesting and worthy of attention.
The paper is well prepared, but I am providing a detailed evaluation to improve it further:
The title of the article is too long and deserves to be shortened.

The abstract is not ideal and the "Highlights" used should be directly included in the context. The abstract contains abbreviations without explanation. It needs to be significantly improved.

The introductory section is extensive and contains a detailed look at chemical behavior. Its scope would have been better divided into subchapters. 
Given the wide range of readers of the Journal, I find the article lacking in providing at least a basic broader context for the justification of your research. For example, some information on the benefits, on the sustainability of the process, on concrete enhancement, etc. For examples, see:
10.1016/j.conbuildmat.2021.125096
10.3390/app11114964
10.1061/(ASCE)MT.1943-5533.0003938

The description of the experimental program and material is logical and understandable. 
I appreciate the chemical description and composition.
The description of the test methods is also in order. 

The results of your research are interesting, but according to the data presented they are as expected.
Figure 2,3,4 lacks a longer direct evaluation - I know you have it in the Discussion, but it is better to include it now. 

Similarly other results - you can't leave everything to the discussion and need to elicit more observations even in words. 

The discussion is extensive, which may be detrimental in the context of the results. But don't take this as a big problem. It is clear that you have thought about the results in detail. 

The conclusions are more of a simulation of the discussion - the conclusions usually do not include further citations and results of other studies. The conclusion seems inadequate and needs to be modified. 

Formal stuff:
Author roles and other mandatory sections are missing,
you have decimal points in figures - should be dots,
for all figures, it would be useful to include a precise description of the symbols and colours in the caption or in the text itself - even if you have it in the form of a legend and abbreviations. 

Author Response

(The authors gave the same response as above.)

Round 2

Reviewer 3 Report

Thank you for the corrections.
Unfortunately, I feel that you have not thoroughly gone through my suggested changes and comments. 
Some are bent, some are ignored without explanation. 

I miss the detailed Response to review where you defend your views point by point.

The numbering of the subchapters does not correspond to the format of the journal.

The current state is not appropriate - please take the time to edit. 
This is a highly ranked journal and needs to be treated as such. 

Author Response

Dear Reviewer,

Thank you for your review of our article and valuable feedback. The first review circuit did indicate our insufficient editing and revision, especially considering the scientific level and quality of the journal. We are corrected our previous response for first round of Review and added new comments. Please find our actions to make your suggestions for improving the article:

First round of Review (corrected answers):

Reviewer: The title of the article is too long and deserves to be shortened.

The title is changed. New shorter title is “Sufractant and calcium sulfate type impact on air entraining effectiveness of concrete”

Reviewer: The abstract is not ideal and the "Highlights" used should be directly included in the context. The abstract contains abbreviations without explanation. It needs to be significantly improved

The abstract was corrected as follows:

Abstract: The paper presents the evaluation of the influence of calcium sulfate on the air void microstructure in concrete and its action mechanism depending on the character of the air-entraining agent. Gypsum dehydration has been previously proven to negatively influence the air void structure of air-entrained concrete. Ettringite, nucleating from tricalcium aluminate and calcium sulfate, influences the adsorption and mode of action of anionic-based polycarboxylate ether admixtures. The authors suspect the admixture’s air-entraining mechanism is also affected by these characteristics. Gypsum dehydration was confirmed to influence the air void structure. In the case of the anionic surfactant, the content of air bubbles smaller than 300 µm was lower compared to cement with gypsum and hemihydrate. On the other hand, the content of air voids with a diameter up to 60 µm, which are the most favorable, was higher. The results obtained lead to the conclusion that the mechanism of air entrainment is twofold and in most cases occurs through the lowering of surface tension and/or through the adsorption of surfactant on cement grains. The adsorptive mechanism was proved to be more effective in terms of the total air content and the structure of the air void system. The results and conclusions of the study give guidelines to determine proper surfactant type, reducing risk of improper air entrainment of concrete and emphasize importance of gypsum dehydration of cement in the process of air entrainment.

Reviewer: The introductory section is extensive and contains a detailed look at chemical behavior. Its scope would have been better divided into subchapters. 

The division into sub-chapters was introduced as follows: Cement hydration, Role of calcium sulfate in cement hydration, Chemical admixtures adsorption mechanisms and Theories of air entraining agents adsorption mechanism. The correct form numbering and formatting was made.

Reviewer: Given the wide range of readers of the Journal, I find the article lacking in providing at least a basic broader context for the justification of your research. For example, some information on the benefits, on the sustainability of the process, on concrete enhancement, etc

The sentence was added to the abstract:

„The results and conclusions of the study give guidelines to determine proper surfactant type, reducing risk of improper air entrainment of concrete and emphasize importance of gypsum dehydration of cement in the process of air entrainment.”

The suggested article examples are interesting and we hope I understand the core of your comment, notwithstanding that the articles mentioned relate to the effect of nano-SiO2 on the pore distribution in concrete, the use of new kind of SCM or the waste Jarosite additive to asphalt. The benefits are evident in case of sustainability. But, the our study focuses primarily on understanding the deterioration of aeration efficiency due to "problems" with cement production (partial dehydration of gypsum in cement during cement grinding or storage), so it is aimed mainly at specialists in the production of cement and using cement in the production of concrete. Of course, it is also profitable for environment as consequence of reduction the negative effect - but in our opinion, highlighting this side in this case would be a bit out of place.

Reviewer: The results of your research are interesting, but according to the data presented they are as expected.

Yes, because the results are real however there are a couple of conclusions giving in our opinion new information to the current state of science:

  1. Anionic surfactant mode of operation is mostly based on adsorption on solid body mechanism , meanwhile cationic on Surface tension reduction between air and liquid
  2. Anionic surfactants are more effective, than cationic
  3. Frost resistance anticipation, based only on A300 is not necessarily the most rightful,, because content of smaller air bubbles is more critical in this context
  4. Obtained results provoke next steps of the research, in order to provide practical guidelines for cement producers, authorities impacting definition of regulations and concret/admixture technologists

Reviewer: Figure 2,3,4 lacks a longer direct evaluation - I know you have it in the Discussion, but it is better to include it now. Similarly other results - you can't leave everything to the discussion and need to elicit more observations even in words. 

Yes, indeed. We add comments under the figs 2-4 as follows:

Figure 2. In the case of anionic surfactant, the impact of calcium sulfate type is moderate, lower on spacing factor and higher on A300.  Gypsum provides the best air microstructure, due to lowest spacing factor and highest A300. Hemihydrate and anhydrite bring similar air system. 

Figure 3. In the case of non-ionic surfactant, the impact of calcium sulfate type is high, both on spacing factor and on A300.  Gypsum provides the best air microstructure, due to lowest spacing factor and highest A300, then hemihydrate and anhydrite the worst.

Figure 4. In the case of cationic surfactant, the impact of calcium sulfate type is high, both on spacing factor and on A300.  Gypsum provides the worse air microstructure, due to highest spacing factor and lowest A300.

Additionally we add consequently one comment under fig 8. The TOC in the case of anionic surfactant AEA-1 is the lowest, what shows high level of adsorption, meanwhile the surface tension is comparable to solution with no AEA. TOC of non-ionic AEA-2 and cationic AEA-3 surfactants is high, what show low level of adsorption, meanwhile for AEA-2 the surface tension is significantly reduced.

Reviewer: The conclusions are more of a simulation of the discussion - the conclusions usually do not include further citations and results of other studies. The conclusion seems inadequate and needs to be modified. 

Conclusions are modified:

  • The results of pore solution analysis proved that air entrainment in concrete can be achieved through one of two mechanisms: the adsorption of anionic surfactant on the ettringite surface or the lowering of surface tension.
  • The anionic surfactant was characterized by a better foamability and provided a better structure of the air void system; thus, the adsorptive mechanism of air entrainment can be pointed out as the more effective.
  • Gypsum dehydration has been proven to alter the air void structure of concrete. Due to gypsum dehydration, the content of air voids with a diameter up to 300 µm decreased by approx. 18% when the anionic surfactant was used. However, the content of air voids with a diameter up to 60 µm, which are the most favorable in terms of the freeze-thaw durability.
  • Gypsum dehydration influenced the air void structure by altering the hydration of C3A and the formation of ettringite rather than affecting the surfactant’s mode of action directly.

Reviewer: Formal stuff:
Author roles and other mandatory sections are missing

The roles of authors are added to the text of paper.

you have decimal points in figures - should be dots.

The decimal points were corrected to the proper format.

for all figures, it would be useful to include a precise description of the symbols and colours in the caption or in the text itself - even if you have it in the form of a legend and abbreviations. 

Information about abbreviations in the drawings (e.g. CGG, CGH and CGA) has been added to the description as follows:

“(CGG – cement with high gypsum content, GDH = 0.11; CGH – cement high hemihydrate content, GDH = 0.37; CGA – cement with anhydrite addition, GDH = 0.17).”

The description is now much clearer for readers.

Also text was added to the paragraph 2.2 Material:

“Used abbreviations of cement type mean in case of CGG – cement with predominant gypsum content, CGH – cement with a significant amount of hemihydrate, and CGA – cement with the addition of anhydrite.”

Second round of Review:

Your review is very valuable and in our opinion, it contributed to a better form of work that we hope meets the high requirements of the journal. Also in our opinion now we add a more precise response to review in form point by point. We corrected (we hope) all of the mistakes about the format and style of the journal.